# Instant4D: 4D Gaussian Splatting in Minutes

**Zhanpeng Luo**
University of Pittsburgh
ZhanpengLuo@pitt.edu

**Haoxi Ran**∗
Carnegie Mellon University
ranhaoxi@cmu.edu

**Li Lu**
Sichuan Univeristy
luli@scu.edu.cn

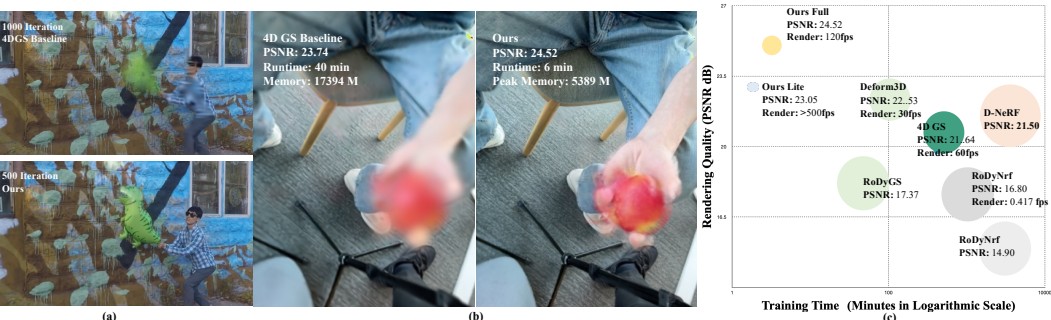

Figure 1: **Part (a):** INSTANT4D achieves better rendering performance with fewer training iterations against the original 4D Gaussian Splatting (4DGS) [41]. **Part (b):** Visualization on detailed dynamic object like a "spinning" apple. After 40-minute optimization, the rendering result of 4DGS remains blurry, while our method achieves better visual quality by 0.8 dB PSNR, faster optimization for convergence by 85%, and lower GPU memory by 69%. **Part (c):** Bubble chart comparing with most recent art. Note that the bubble size indicates the size of an optimized model.

## Abstract

Dynamic view synthesis has seen significant advances, yet reconstructing scenes from uncalibrated, casual video remains challenging due to slow optimization and complex parameter estimation. In this work, we present INSTANT4D, a monocular reconstruction system that leverages native 4D representation to efficiently process casual video sequences within minutes, without calibrated cameras or depth sensors. Our method begins with geometric recovery through deep visual SLAM, followed by grid pruning to optimize scene representation. Our design significantly reduces redundancy while maintaining geometric integrity, cutting model size to under **10%** of its original footprint. To handle temporal dynamics efficiently, we introduce a streamlined 4D Gaussian representation, achieving a **30×** speed-up and reducing training time to within two minutes, while maintaining competitive performance across several benchmarks. Our method reconstruct a single video within 10 minutes on the Dycheck dataset or for a typical 200-frame video. We further apply our model to in-the-wild videos, showcasing its generalizability. Our project website is published at https://instant4d.github.io/.

## 1 Introduction

Reconstructing dynamic 3D scenes from casually captured, uncalibrated video is a fundamental challenge in computer vision, critical for applications such as augmented reality (AR), virtual reality (VR), and immersive content creation. While static 3D scene modeling has seen remarkable

---

∗Project Lead

39th Conference on Neural Information Processing Systems (NeurIPS 2025).

progress [13, 1, 22, 21, 23, 31], extending these techniques to dynamic scene remains challenging, especially when handling moving objects with monocular camera only. This process often requires time-consuming optimization [12, 3, 26] to recover scene geometry and accurate motion. Furthermore, occlusion, deformation, and irregular camera paths add complexity, making efficient and coherent modeling difficult in uncalibrated settings.

Recent approaches leverage optical flow [12], depth [9], point-tracking [29], and pose prior [5] to solve this challenging task. Nevertheless, reconstructing from a short, causal video still requires hours of optimization. Inspired by recent advances in deep visual SLAM [11] and real-time rendering [6, 41] we propose INSTANT4D, a reconstruction system for dynamic scene reconstruction in only minutes. We employ deep visual SLAM to estimate camera trajectories and refine the monocular depth into video consistent depth. These depth maps are then back-projected into a dense 3D point cloud as 4DGS optimization. Furthermore, we propose a grid pruning strategy, which efficiently reduces redundancy while preserving occlusion structures, reduces the model size to less than **10%** of its original footprint and significantly accelerates the optimization process, achieving **30×** acceleration compared to recent works of art.

Then, we model the attributes of dynamic scenes with the native 4D Gaussian primitive [40, 41] that captures motion without rigidly segmenting the scene into static and dynamic parts. Unlike previous approaches [12, 5, 9, 29, 33], our method further enables naturally captures on some background variations. However, modeling sparse and temporally inconsistent observations make the 4D Gaussian overfit and prematurely disappear in poorly observed regions. We address this through a carefully crafted initialization scheme and a motion-aware 4D covariance model.

INSTANT4D demonstrates short training time, low peak memory, fast rendering speed, and high rendering quality, as shown in Figure 1. Specifically, we reconstruct scenes on NVIDIA dataset [43] in average 2 minutes and on Dycheck [3] dataset in average 7.2 minutes. For a typical 5-second, 30 FPS video, our method completes optimization within 8 minutes. We achieve 30× speed-up in reconstruction time. 90% reduction in memory and demonstrate competitive performance on several benchmarks. Our primary contributions are summarized as follows.

- We propose INSTANT4D, a modern and fully automated pipeline that reconstructs casual monocular videos within minutes, achieving **30×** speed up.

- We introduce a grid pruning strategy that reduces the number of Gaussians by **92%**, preserving the occlusion structures and enabling scalability to long video sequences.

- We present a simplified, isotropic, motion-aware 4DGS formulation, in monocular setup, which achieves **29%** better performance than current state-of-the-art methods on the Dycheck dataset.

## 2 Related Work

### 2.1 Dynamic Novel View Synthesis (NVS)

**NeRF-based NVS** Earlier methods like [43] used single view and multiview stereo depth to synthesize novel views of dynamic scenes from a single video using explicit depth-based 3D warping. A recent line of work [20, 12, 16] extends NeRF [14] to handle a dynamic scene by adding a time dimension. In particular, RoDynRF [12] split the scene into static and dynamic parts and used the static radiance field [14] to estimate only the camera poses, which were robustly reconstruct from unposed RGB video. However, limited by Neural Radiance Fields' rendering speed and numerous iteration demands, usually RoDynRF [12] takes over 2 days to reconstruct a casual video.

**Gaussian-based NVS** Approaches to modeling motion with Gaussian Splatting can be broadly categorized into three types: *deformation-based*, *trajectory-based*, and *4D-Gaussian-based* methods. Deformation-based methods [35, 42, 36, 7] employ multi-layer perceptrons (MLPs) or low-rank K-planes to dynamically adjust the parameters of Gaussians over time. Although expressive, these methods typically suffer from slower training and rendering due to the added complexity of learning continuous deformations. Trajectory-based methods [29, 9, 28] explicitly model the motion of Gaussians by pre-computing trajectories, often derived from external motion estimators. Although this enables precise tracking of object movement, it demands substantial preprocessing. For example,

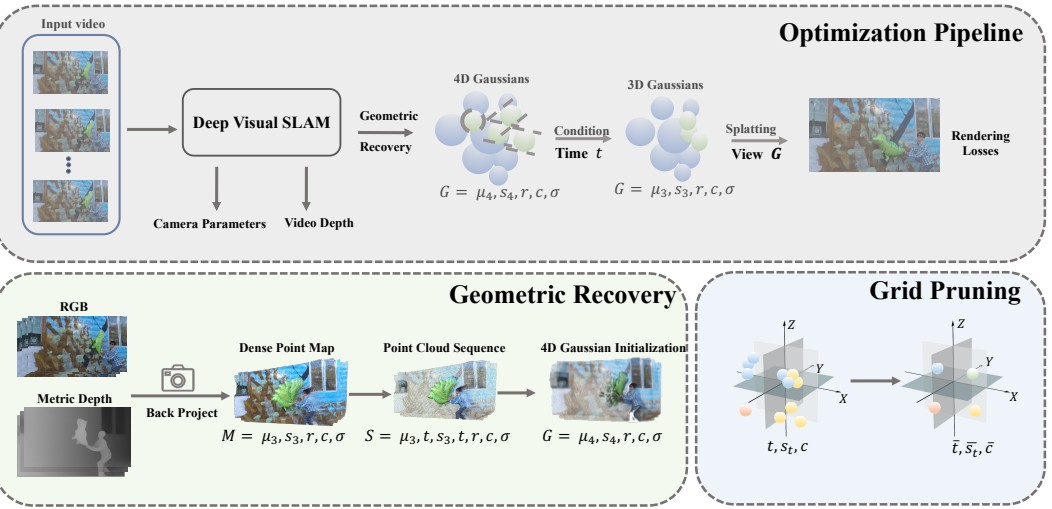

Figure 2: Pipeline of INSTANT4D. We use Deep Visual SLAM model and Unidepth [19] to obtain camera parameters, and metric depth. The metrics depth would be further optimized to consistent video depth. After that we back project from consistent depth to get dense point cloud, further voxel filtered to sparse point cloud, as discuss in Section 3.2. Based on the 4d Gaussians Initialization, we can reconstruct a scene in 2 minutes. More details about optimization are described in Section 3.3.

a 3-second video can generate trajectory files of more than 100 GB [29]. Furthermore, these methods rigidly partition the scene into static and dynamic components, neglecting subtle background motion that may still be perceptible. This hard segmentation introduces artifacts when background elements exhibit slight temporal shifts. Finally, 4D-Gaussian-based methods [38, 41, 37] extend the standard 3D Gaussian Splatting by including a temporal dimension to the representation. At a given timestamp, the 4D Gaussian is conditioned to a 3D distribution. However, it is prone to overfitting in monocular settings where certain regions are briefly visible only; without careful temporal management, 4D Gaussians tend to vanish prematurely as they are underconstrained in time.

## 2.2 Visual SLAM and SfM

Classical structure-from-motion (SfM) and Simultaneous Localization and Mapping (SLAM) pipelines recover camera poses and sparse geometry by minimizing re-projection or photometric errors through bundle adjustment [4, 24]. Deep visual SLAM systems, such as DROID-SLAM [27] replace handcrafted heuristics with differentiable bundle adjustment layers and data-driven priors, resulting in improved robustness in texture-poor scenes and mild dynamics. Recently, MegaSAM [11] has extended the differentiable bundle adjustment to dynamic scenes. MegaSAM leverages data learned before camera and flow supervision, robustly recovers camera parameters, and generates consistent video depth. Another notable recent data-driven method, DUSt3R [32], powered by a CroCo encoder [34], learned from vast pretrained data, reconstructs camera poses quickly and robustly. Follow efforts such as MonST3R [44] extend DUSt3R to a dynamic scene with additional dynamic supervision. CUT3R [30] also applies the CroCo [34] encoder and applies continuous learning for static and dynamic reconstruction.

## 3 Method

Given an unconstrained video sequence $\mathcal{V} = \{I_i\}_{i=1}^N$ with resolution $H \times W$, our goal is to estimate camera extrinsics $\hat{\mathbf{G}}_i \in \mathrm{SE}(3)$, intrinsics $K \in \mathbb{R}^{3 \times 3}$, and temporally consistent depth maps $\hat{D} = \{\hat{D}_i\}_{i=1}^N$. Leveraging these estimates, we reconstruct a dynamic 4D scene representation and render novel views in real-time from arbitrary viewpoints $\mathbf{G}^*$ at given timestamps $t^*$ using Gaussian Splatting. Our pipeline is illustrated in Figure 2 We first briefly summarize the relevant background on

deep visual SLAM and Gaussian Splatting (Section 3.1), before detailing our geometric initialization pipeline (Section 3.2) and optimization strategy (Section 3.3).

## 3.1 Preliminary

**MegaSAM** [11] extends the differentiable bundle adjustment framework from DROID-SLAM [27] to handle dynamic monocular videos. Specifically, (initialized by Depth Anything [39]), camera poses $\hat{\mathbf{G}}_i \in \mathrm{SE}(3)$, and camera intrinsics represented by focal length $f$ (initialized by Unidepth [19]).

During optimization, MegaSAM jointly refines these parameters by iteratively minimizing the weighted reprojection residuals between predicted optical flow and rigidly computed flow from current estimates:

$$\mathbf{u}_{ij} = \pi\left(\hat{\mathbf{G}}_{ij} \circ \pi^{-1}(\mathbf{p}_i, \hat{\mathbf{d}}_i, K^{-1}), K\right), \tag{1}$$

where $\hat{\mathbf{G}}_{ij}$ denotes the relative transformation from frame $i$ to frame $j$, and $\pi(\cdot)$ denotes the camera projection operation.

MegaSAM optimizes these parameters using the Levenberg–Marquardt (LM) algorithm:

$$\left(\mathbf{J}^\top \mathbf{W} \mathbf{J} + \lambda \operatorname{diag}(\mathbf{J}^\top \mathbf{W} \mathbf{J})\right) \Delta = \mathbf{J}^\top \mathbf{W} \mathbf{r}, \tag{2}$$

where $\Delta = (\Delta\mathbf{G}, \Delta\mathbf{d}, \Delta f)^\top$ is the parameter update, $\mathbf{J}$ is the Jacobian of reprojection residuals $\mathbf{r}$ with respect to the parameters, and $\mathbf{W}$ is a diagonal weighting matrix derived from each frame pair. The damping factor $\lambda$ is adaptively predicted by the network during each iteration to stabilize optimization.

**4D Gaussian Splatting** [41] extends the explicit 3D Gaussian Splatting [6] to dynamic scenes by incorporating temporal dynamics into scene modeling. Specifically, a standard 3D Gaussian is parameterized by its mean position $\boldsymbol{\mu} \in \mathbb{R}^3$, covariance matrix $\Sigma \in \mathbb{R}^{3\times3}$, and opacity $\alpha \in \mathbb{R}$ as follows:

$$G(\mathbf{p}, \boldsymbol{\mu}, \Sigma, \alpha) = \alpha \exp\left(-\frac{1}{2}(\mathbf{p} - \boldsymbol{\mu})^\top \Sigma^{-1}(\mathbf{p} - \boldsymbol{\mu})\right). \tag{3}$$

To represent a view- and time-dependent appearance, 4DGS employs a set of 4D sphericylindrical harmonics (SCH), constructed by combining 3D spherical harmonics (SH) with temporal Fourier basis functions:

$$Z_{nl}^m(t, \theta, \phi) = \cos\left(\frac{2\pi n}{T} t\right) Y_l^m(\theta, \phi), \tag{4}$$

where $Y_l^m$ are the standard 3D spherical harmonics indexed by degree $l \geq 0$ and order $m$ with $-l \leq m \leq l$, $n$ is the temporal frequency index, and $T$ denotes the temporal period.

## 3.2 Geometric Recovery

**Back Projection on Consistent Depth**  Given the input image sequence $\mathcal{V}$, we first apply MegaSAM [11] to obtain estimates of camera extrinsics $\hat{\mathbf{G}}_i \in \mathrm{SE}(3)$ and intrinsics $K \in \mathbb{R}^{3\times3}$. We then refine the initial monocular depth estimates to achieve temporally consistent depth maps $\hat{D} = \{\hat{D}_i\}_{i=1}^N$ through an additional first-order optimization. Using these refined depths, we back-project each pixel coordinate $\mathbf{p}_i$ from the image space into 3D world coordinates $\mathbf{X}_i \in \mathbb{R}^3$:

$$\mathbf{X}_i = \hat{\mathbf{G}}_i \circ \pi^{-1}(\mathbf{p}_i, \hat{D}_i, K^{-1}), \tag{5}$$

where $\pi^{-1}$ denotes the back-projection of pixel $\mathbf{p}_i$ (in homogeneous coordinates $\tilde{\mathbf{p}}_i$) and depth $\hat{D}_i$ into the camera frame, and $\hat{\mathbf{G}}_i$ transforms the 3D point into the world coordinate frame. This procedure yields a dense colored point cloud representing the scene geometry. To handle variations in depth scale, particularly in outdoor scenes with unbounded regions (e.g., skies), we adaptively increase the voxel size $S_v$ during subsequent grid pruning.

**Motion Probability Estimation**   Separating dynamic foreground objects from the static background remains beneficial, as it allows us to efficiently allocate computational resources by representing the static background sparsely while preserving detailed granularity in dynamic regions. To this end, we leverage intermediate predictions from the deep visual SLAM pipeline's low resolution motion probability map $\hat{m} \in \mathbb{R}^{\frac{H}{8} \times \frac{W}{8}}$. to interpolate to a per-pixel motion probabilities. We then employ Otsu's thresholding method [15] on these probability maps to generate binary masks that distinguish static from dynamic scene elements. Empirically, we observed that in sequences with large temporal sampling intervals, motion estimation can fail to reliably identify moving objects at the sequence boundaries (i.e., first and last frames). To mitigate this issue, we introduce synthetic pseudo-frames at both ends of the sequence, thereby improving motion consistency. Additional visualizations of our motion estimation procedure are provided in the supplementary materials.

**Grid Pruning**   Back-projecting depth maps for a $512 \times 512$ video sequence of four seconds (30 FPS) yields $\sim 30\,$M raw 3D points. To eliminate redundancy and resolve self-occlusions, we partition the world space into a regular voxel grid and retain only the centroid of points within each occupied voxel. The edge length is adapted to the scene scale,

$$S_v = \lambda_s \cdot \frac{1}{N} \sum_{i=1}^{N} \frac{\hat{D}_i}{\hat{f}}, \tag{6}$$

where $\hat{D}_i$ is the mean depth of the frame $i$, $\hat{f}$ the estimated focal length, $N$ the number of frames, and $\lambda_s$ a user-defined scale factor. Each 3D point is assigned to a voxel by integer division with $S_v$; points in the same cell are aggregated via a hash-map and replaced by their centroid, while other attributes other than position, such as color, timestamp, time scale $s_t$, and motion probability $\hat{m}$ are averaged. Voxels with insufficient support are discarded as outliers to suppress noise.

In the NVIIDA Dynamic Scene benchmark [43], this pruning reduces the model's memory footprint from 10.7 GB to 0.83 GB (92%), reduces the training time from 181 s to 42 s (4× speed-up), and improves rendering performance from 154 FPS to 981 FPS (see Table 2). With the resulting compact geometric priors, we can reduce the need of the densification stage conventionally applied by 3D Gaussian Splatting [6].

### 3.3   Optimization

**Motion Modeling**   Once 3D positions and RGB colors are acquired, we optimize the remaining Gaussian attributes such as rotation $r$, scaling $s$, opacity $o$, and motion $m$ with a lightweight 4D Gaussian formulation. Each Gaussian is described by a 4D mean $\boldsymbol{\mu} = (\mu_x, \mu_y, \mu_z, \mu_t)^\top \in \mathbb{R}^4$, a diagonal scale vector $\mathbf{s} = (s_{xyz}, s_t)^\top$, a scalar opacity $\alpha$, and a rotation matrix $R \in \mathbb{R}^{4 \times 4}$. Unlike prior 4DGS [41] work that relies on two entangled quaternions and high-order SCH, we model the Gaussian's appearance with simple RGB value, rather than a high-order spherical harmonious function. The simple design cuts the per-Gaussian parameter count by over 60% and empirically lessens over-fitting in monocular settings.

We condition a multivariate 4D Gaussian primitive towards a 3D Gaussian primitive at timestamp $t$ during the rendering time, which can be formulated as follows:

$$\boldsymbol{\mu}_{xyz|t} = \boldsymbol{\mu}_{1:3} + \Sigma_{1:3,4}\,\Sigma_{4,4}^{-1}\,(t - \mu_4), \tag{7}$$

$$\Sigma_{xyz|t} = \Sigma_{1:3,1:3} - \Sigma_{1:3,4}\,\Sigma_{4,4}^{-1}\,\Sigma_{4,1:3}, \tag{8}$$

where $\Sigma \in \mathbb{R}^{4 \times 4}$ is the full covariance matrix. This conditioned form encodes continuous motion without explicit trajectory storage.

**Isotropic Gaussian**   Although anisotropic Gaussians can model fine-grained shape details, their additional degrees of freedom frequently destabilize optimization in monocular scenarios. Inspired by Gaussian Marbles [25], we therefore adopt an *isotropic* variant: the orientation matrix is fixed to the identity ($R = I$), and the covariance is parameterized by two scalars, one shared spatial scale $s_{xyz}$ and one temporal scale $s_t$. This compact parameterization improves numerical stability, reduces memory usage, and acts as an implicit regularizer. As evidenced in Table 4, the isotropic model delivers higher robustness without sacrificing rendering quality.

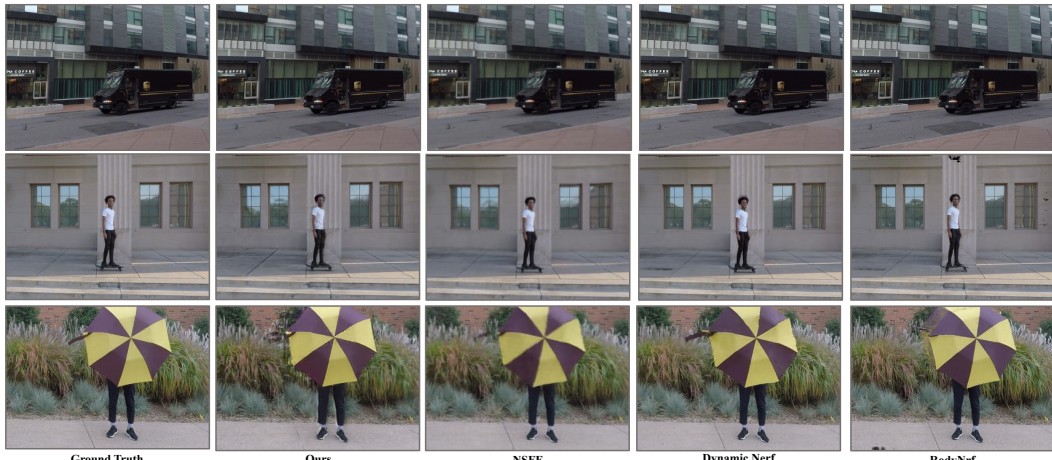

Figure 3: Visual comparison on the NVIDIA dataset. [43]

| Method | Calibration | Runtime ↓ | Rendering FPS ↑ | | PSNR ↑ |
| --- | --- | --- | --- | --- | --- |
| | | | $480 \times 270$ | $860 \times 480$ | |
| HyperNeRF [17] | COLMAP | 64 | 0.40 | - | 17.60 |
| DynamicNeRF [3] | COLMAP | 74 | 0.05 | - | **26.10** |
| RoDynRF [12] | COLMAP | 28 | 0.42 | 0.13 | 25.89 |
| 4DGS [35] | COLMAP | 1.2 | 43 | 29 | 21.45 |
| Casual-FVS [8] | Video-Depth-Pose | 0.25 | 48 | 27 | 24.57 |
| InstantSplat* [2] | Visual SLAM | 0.15 | 117 | - | 22.56 |
| 4DGS* [41] | Visual SLAM | 0.16 | 98 | - | 18.34 |
| **Ours** | Visual SLAM | **0.02** | **822** | **676** | 23.99 |

Table 1: Quantitative comparison of efficiency and visual quality on NVIDIA dataset following [12]. ∗: Our implementation on the same server by replacing the calibration method (COLMAP) from the original paper with Visual SLAM for fair comparison.

**Motion-Aware Gaussian**    In monocular 4DGS primitive modeling, static background primitives can vanish once they leave the camera frustum unless they are explicitly distinguished from moving objects. We apply the mask obtained from 3.2.

To make our 4D primitive aware of underlying motion in the monocular dynamic scene. Considering opacity $o_t = o \times \mathcal{N}(t, \mu_4, \Sigma_{4,4})$ and Equation 7, we can find that in the case of our isotropic Gaussians, the temporal scaling $s_t$ would be the only term in the covariance affect the Gaussian attribute related with time.

$$\Sigma_{4,4} = s_t \times s_t \tag{9}$$

Therefore, by explicitly setting $s_t$ higher for the static region, those Gaussians that should remain in the 4D space will not disappear. Those dynamic Gaussians will change their position and scaling according to the deviation of timestamp $t$. During rendering, Gaussians farther away from the timestamp $t$ will be culled if their opacity $o_t = o\mathcal{N}(t; \mu_4, \Sigma_{4,4})$ falls below a threshold.

## 4 Experiments

### 4.1 Training and Inference Detail

**Implementation Detail**    On the Dycheck iPhone dataset [3], we followed the evaluation protocol established by Jeong et al [5]. We set the maximum optimization iterations to 5,000 and adopted the standard 3DGS[6] hyperparameters for loss weights and learning rates, with the exception of reducing the position learning rate to $1e^{-5}$ and extending the learning rate scheduler to 5,000 steps.

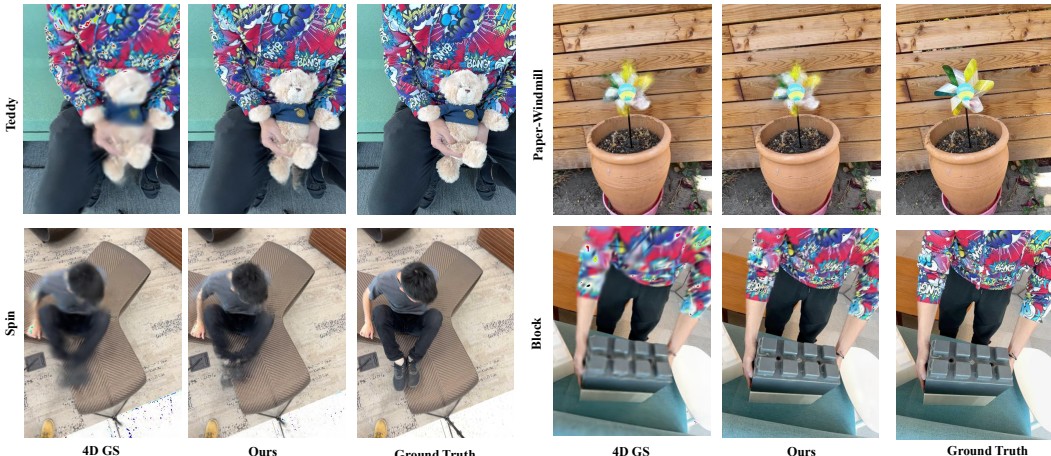

Figure 4: Visual Comparison on the Dycheck dataset.[3]

| Component | SH | Filter | Densification | PSNR ↑ | Runtime (sec) | Memory (MB) | Rendering FPS |
|---|---|---|---|---|---|---|---|
| **Ours** | ✓ | ✓ | | **23.99** | **42** | **832** | **981** |
| SCH | | ✓ | | 23.83 | 59 | 2806 | 588 |
| W.o. Voxel Filter | | | ✓ | 23.38 | 181 | 10676 | 154 |

Table 2: Ablation study on key components of INSTANT4D's influence on speed and memory. We analyze the effect of spherical harmonics (SH), grid filtering, and Gaussian densification on rendering quality, training runtime, memory usage, and rendering frame rate (FPS). Removing higher-order SCH 3.1 slightly reduces computational cost with minimal impact on PSNR. The grid filtering significantly reduces both memory footprint and runtime while maintaining rendering quality, highlighting its role as an effective regularizer against overfitting.

Our initialization strategy differed between model variants. For the *Lite* model, we initialized 4D Gaussians with a voxel size of $\lambda_s = 4$ for static regions and $\lambda_d = 4$ for dynamic regions. In our *Full* model, we set $\lambda_s = 1$ but omitted the grid pruning step for dynamic regions to preserve detail and alleviate some potential underfit caused without densification. During practice, one can still enable densification for better training view rendering quality. Temporal scaling was set to $s_t = \frac{k}{fps}$ for dynamic regions, while static regions used a constant scale equal to the entire video length ($s_t = l_{video}$).

For the NVIDIA Dynamic Scene dataset [43], we reduced the maximum optimization iterations to 1,500 while maintaining the same hyperparameters as our implementation in Dycheck [3], adjusting only the learning rate scheduler's maximum step to match the shorter optimization cycle. The grid pruning and initialization parameters remained consistent across both datasets.

**Runtime and Memory** The computational requirements for our method scale with input video length, as the SLAM system must track additional depth maps. Peak memory usage occurs during consistent video depth optimization, while 4DGS optimization maintains relatively stable runtime regardless of sequence length. Testing on a single NVIDIA A6000 GPU, our Lite model completes the full training pipeline in 96 seconds with peak memory usage of 988 MB on the shortest sequence (235-frame "paper-windmill"), and 131 seconds with peak memory of 1,147 MB on the longest sequence (379-frame "apple"). For our Full model, geometric recovery processes at approximately 0.8 seconds per frame, requiring about 5 minutes total for depth estimation, video depth consistency optimization, and camera tracking on the "apple" sequence. Inference runs at over 400 Hz, and the voxel pruning stage completes in less than 5 seconds per scene.

## 4.2 Evaluation on NVIDIA & Dycheck Benchmarks

**Evaluation on NVIDIA** We evaluated INSTANT4D against several baseline methods in the NVIDIA Dynamic data set following the protocol [12]. This dataset consists of seven scenes, each with 12

| PSNR(↑) | Apple | Block | Paper | Spin | Teddy | Average | Runtime(h) | Mem(GB) |
|---|---|---|---|---|---|---|---|---|
| D-NeRF [20] | 24.23 | 21.80 | 21.85 | 22.15 | 19.46 | 21.50 | > 24 | 12 |
| RoDynRF [12] | 17.38 | 15.99 | 20.71 | 16.66 | 13.28 | 16.80 | 22 | 15 |
| 4DGS [35] | 23.24 | 22.05 | 21.03 | 22.99 | 18.89 | 21.64 | 1.2 | 21 |
| Deform3D [42] | 24.82 | 23.26 | 20.62 | 23.51 | 20.93 | 22.63 | - | - |
| RoDynRF [12] (w.o. pose) | 14.50 | 14.73 | 17.94 | 15.75 | 11.56 | 14.90 | 22 | 15 |
| RoDyGS [5] | 16.79 | 17.67 | 19.20 | 18.47 | 14.69 | 17.37 | 1.0 | - |
| **Ours** (Lite) | 24.9 | 23.48 | 23.18 | 23.60 | 19.96 | 23.02 | **0.03** | **1.1** |
| **Ours** (Full) | **26.84** | **23.98** | **24.77** | **25.25** | **21.78** | **24.52** | 0.12 | 8 |

Table 3: DyCheck iPhone benchmark [3]. Methods above the mid-rule are trained with ground-truth camera; those below operate without calibrated poses. *Runtime* denotes the mean training time per scene and *Mem* the peak GPU memory during optimization. Runtime for RoDyGS, RoDynRF, and D-NeRF is provided by the authors of [5].

frames captured from 12 camera viewpoints for training, with testing performed from fixed viewpoints at consecutive timestamps. The visualization is shown in Figure 3.

To isolate the contributions of our approach, we developed two comparative baselines with similar training time constraints. The first **InstantSplat [2] style baseline** , adapt Fan et al.'s approach [2], with covisible global geometry initialization and joint camera pose optimization. For the previous part, we tested with counterparts against MAST3R [10] such as CUT3R [30] and MonST3R [44], but these models struggle with frequently shifting point clouds when processing a long sequence. Therefore, we still use MegaSAM [11] as the visual SLAM model. As shown in Table 1, our model achieves a higher rendering quality while maintaining significantly faster training times, demonstrating the effectiveness of our grid pruning and initialization strategy.

Furthermore, we introduce a 4DGS [41] baseline, This baseline isolates the contribution of our 4D Gaussian representation by implementing a standard 4DGS approach without our isotropic and motion-aware Gaussian strategies. As shown in Table 1, our full model achieves superior rendering quality while maintaining significantly faster training times, demonstrating the effectiveness of our grid pruning and initialization strategy. The ablation results in Table 4 further confirm that omitting motion-aware Gaussians substantially degrades the quality of rendering across both datasets. This degradation likely stems from overlapping dynamic elements in world space, where improperly timestamped Gaussians occlude each other and impede optimization.

While some prior methods such as [12, 3] achieve higher PSNR values, they typically incorporate additional regularization techniques to compensate for the limited information available in the 12-frame NVIDIA dataset. Nevertheless, our method offers a compelling trade-off, delivering competitive quality with reconstruction speeds that dramatically outpace previous approaches.

**Evaluation on DyCheck** The DyCheck iPhone benchmark [3] presents severe motion and parallax, making it a stringent test for dynamic reconstruction. Following the RoDyGS [5] evaluation protocol, we report PSNR per-scene together with training time and peak memory (Table 3. Our *Lite* variant already surpasses all baseline methods that do not require ground truth camera poses as input, achieving an average **23.02 dB** in just **0.03 h** with a footprint of 1.1 GB. The *Full* configuration lifts performance to **24.52 dB**, outperforming the concurrent RoDyGS by **7.15 dB** and exceeding Deform3D[42] (which uses ground-truth poses) by **1.89 dB**, yet still trains in only 7.2 minutes.

Both variants maintain real-time rendering (>500 FPS), confirming that our voxel-initialized, simplified 4D Gaussian representation delivers state-of-the-art quality at a fraction of previous computational cost.

From the visual comparison Figure 4, we can see that compared to the baseline 4DGS model, our method preserves significantly better for static background as well as for the dynamic object.

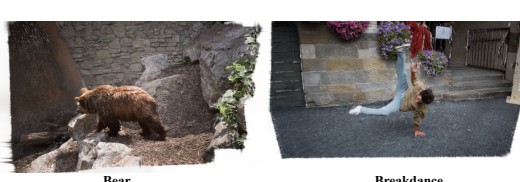

Figure 5: Visualization on the DAVIS Dataset.

## 4.3 Evaluation on in-the-wild video

To assess performance on in-the-wild video, we conduct qualitative experiments on DAVIS Dataset [18]. Figure 5 shows renderings from both novel viewpoints and novel timestamps; complete video results are available on our project website. Our reconstructions exhibit crisp object boundaries and temporally coherent appearance. For example, both the *Bear* sequence (82 frames) and *Breakdance* sequence (68 frames) require 2 min for SLAM calibration and 2 min for 4D reconstruction.

Additionally, we further discuss failure cases. In the low-texture *Kite-surf* sequence, the ocean dominates the field of view, leading to inaccurate visual-SLAM poses; consequently, the surfer occasionally disappears in the rendered output. In our future work, we tend to address such degenerate scenarios for texture-robust pose initialization.

## 4.4 Ablation and Analysis

We first conduct experiments on the NVIDIA [43] dataset to evaluate the impact of each component on training runtime and memory as seen in Table 2. The grid pruning significantly reduces both memory footprint and runtime while maintaining rendering quality, and the simple RGB value we used shows a beneficial trade-off on both performance and training speed.

To assess the impact of each design choice, we conduct an ablation study on the DyCheck iPhone dataset [3] at $2\times$ resolution (Table 4). Starting from our *Full* model, we disable individual components to evaluate their influence on rendering quality and temporal consistency. First, we replace the per-Gaussian RGB coefficient with the original time-varying 4D sphericylindrical harmonic basis as used in 4DGS. While this configuration increases the parameter count, it provides no benefit in rendering quality and in fact decreases PSNR by **1.0 dB.** This suggests that the simplified RGB representation is not only more efficient, but also better suited for monocular, unconstrained video.

Additionally, we evaluate the impact of our proposed design of Isotropic Gaussian. Typically, 3D Gaussian Splatting methods model the scaling and rotation of an 3D Gaussian primitive with anisotropic covariance. However, we adopt a fixed identity orientation and a scalar for 3D scaling. We find that the reduced flexibility introduces stability, resulting in a **1.25 dB** gain in PSNR. Finally, we investigate the effect of disabling the motion-aware Gaussian strategy, setting the temporal scale uniformly for all Gaussians.

| Component | PSNR ↑ | SSIM ↑ |
|---|---|---|
| **Ours** (Full) | **24.52** | **0.834** |
| w/o Motion Aware Gaussian | 21.11 | 0.721 |
| w/o Isotropic Gaussian | 23.00 | 0.661 |
| w/o zero-degree SH | 23.52 | 0.755 |

Table 4: Ablation study on each component's effective on performance.

This configuration fails to differentiate between static and dynamic regions, leading to motion blur and a substantial reduction in PSNR by **3.4 dB**. The results demonstrate that each component of INSTANT4D, including the compact RGB representation, isotropic Gaussian formulation, and motion-aware temporal scaling, plays a crucial role in maintaining visual fidelity and temporal stability.

## 5 Discussion & Conclusion

**Discussion** While INSTANT4D achieves state-of-the-art efficiency and reconstruction quality, it is currently limited in its scalability to long-duration video sequences. The visual SLAM component retains depth maps for each frame, leading to a linear increase in memory consumption as the sequence length grows. This constraint hinders the application of our method to extended captures, such as multi-minute scenes or continuous video streams. Addressing this bottleneck requires innovations in hierarchical memory management and online depth-map compression, which we consider promising avenues for future research. Furthermore, handling scenes with highly reflective or transparent surfaces remains a challenge, as depth estimation becomes less stable under such conditions.

**Conclusion** We present INSTANT4D, a novel system for fast 4D reconstruction from casual, uncalibrated monocular video. Our approach leverages grid pruning, motion-aware Gaussian Splatting, and efficient 4D representation to achieve real-time rendering speed with limited memory overhead. Experiments on several benchmarks demonstrate its superior rendering visual quality and

computational efficiency compared to existing methods. Our future work will focus on extending our framework to video sequences with arbitrary length, improving scalability through hierarchical SLAM representations, and efficient memory management, while also addressing limitations in highly reflective and low-texture scenes.

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
