# OpenReview forum: "Instant4D: 4D Gaussian Splatting in Minutes"
_NeurIPS.cc/2025/Conference — NeurIPS 2025 poster_

### Official Review · Reviewer_EE6C · 2025-06-18

**Clarity:** 4
**Significance:** 2
**Originality:** 3
**Rating:** 5
**Confidence:** 3

**Summary:**

This paper proposes Instant4D, a novel 4D Gaussian Splatting method for a monocular camera NVS. Prior methods require time for optimization of the 4D scene. Instant4D leverages deep visual SLAM to determine the camera pose, utilizes depth information from video to obtain an initial 3D point cloud, and optimizes the 4DGS representation. The proposed grid pruning strategy further reduces the redundancy at the initialization of the 3D point cloud. Proposed method reduces memory usage by 92%, training time by 4 times, and improves FPS from 154 to 981 on the NVIDIA dataset and shows improvement on the DyCheck dataset.

**Questions:**

While the current evaluation is sufficient, is there a plan to add statistics about the result? In the checklist, it is mentioned that statistical significance is tested; however, only the naive metrics are presented. (error bars, confidence interval, statistical significance test)

How much and what kind of advantage does this method have compared to online 4DGS methods, such as 3DGStream? Their training time is significantly shorter than the typical 4DGS as well, while they are not evaluated on the same dataset. More comparisons with 4DGS methods focusing on such efficiency would make the evaluation more solid.

How well does this method perform, given the COLMAP-based camera pose and initialization? Integration of SLAM comes with the camera parameters and the proposed depth-based initialization. Current analysis shows the impact of the 4DGS representation and the pruning method. Since SLAM-assisted initialization is an impactful factor for the method's efficiency, further analysis would be beneficial.

**Ethical Concerns:**

["NO or VERY MINOR ethics concerns only"]

**Final Justification:**

From my side, concerns are addressed and I keep the original rating.

**Limitations:**

yes

**Quality:**

3

**Strengths And Weaknesses:**

Strength:
* Evaluation shows a significant speedup and reduced memory usage compared to the state-of-the-art.
* Proposed pointcloud initialization looks well-adaptable for different 4DGS methods.

Weaknesses:
* Overall framework adapts existing methods with relatively straightforward novel techniques.

---

> ### Author Rebuttal · Authors · 2025-07-28
>
> Thank you very much for your recognition and valuable feedback. We will revise this paper based on your suggestions. Please feel free to point them out if any of our explanations are unclear.
>
> ## **Weakness:**
> ###  **1:Our Pipeline Design**
> Our motivation is rooted in the recent advances in deep visual SLAM systems, which have significantly improved speed [1], robustness [2], and generalization [3] in geometry recovery from casually captured monocular videos. Traditional 4DGS methods initialize from the sparse point cloud calibrated from COLMAP. Our methods show that in the casual video setting, the training speed will be greatly improved by introducing our initialization strategy.
>
> To best our knowledge, our work is one of the first to tightly integrate deep visual SLAM with 4D Gaussian Splatting, bridging the gap between geometry-centric SLAM research and appearance-centric neural rendering. We fully leverage the deep visual SLAM tool, achieving fast, robust, and efficient 4D reconstruction in monocular settings. We believe this direction is valuable for the community, as it provides a strong baseline, enables real-time rendering, and could be used as a dataset to train feedforward novel view synthesis models for casual video. We will open-source our code to further facilitate our research community.
>
> ## **Question:**
> ### **1: More Statistics about the Result**
> We agree that more statistical analysis would strengthen our claims. In the revised manuscript, we will include standard deviation and confidence intervals for the ablation study, and test the statistical significance against baseline methods. We believe these additions will better reflect the consistency and statistical reliability of our improvements.
>
> Here, we provide a detailed ablation study with mean and standard deviation on the Dycheck dataset by repeating for 5 times:
>
> | Component               | PSNR (Mean) ↑ | SSIM (Mean) ↑| PSNR (Std) | SSIM (Std) |
> |------------------------|--------------|---------------|------------|------------|
> | **Ours (Full)**        | 24.46        | 0.84          | 0.27       | 0.06       |
> ||
> | w/o Motion Aware Gaussian  | 21.34        | 0.77          | 0.35       | 0.07       |
> | w/o Isotropic Gaussian     | 23.12        | 0.74          | 0.53       | 0.04       |
> | w/o zero-degree SH                     | 23.67        | 0.78          | 0.30       | 0.10       |
>
> ### **2: Evaluation Compared with Online 4DGS Methods**
> Thank you for providing these useful suggestions. Due to limited time, we are unable to provide the quantitative comparison here. We will properly cite related work and include the comparison with 3DGStream [4] in our final version.
>
> For qualitative comparison, one advantage of our methods is the initialization strategy. 3DGStream [4] requires a 5000-iteration initialization for the first frame, while in our method all Gaussians will receive gradient updates at the beginning. Besides, our other advantage is the native 4D Gaussian representation, which does not require per-frame neural transformation.
>
> ### **3: Evaluation with COLMAP Initialization & Analysis**
> Here is the experiment evaluated on the NVIDIA dataset with COLMAP-based camera pose:
>
> | Method      | Calibration | Runtime (s) | FPS (480p) | PSNR ↑ |
> |-------------|-------------|-------------|------------|---------|
> | Ours-COLMAP | COLMAP      | 252 s       | 103        | 23.45   |
> | **Ours**    | Visual SLAM | 72 s        | **822**    | **23.99** |
> ----
>
> Note that for our method initialized from COLMAP, we turn off the motion-aware Gaussian design, enable densification, and set the number of iterations to 5000. For the **Ours** runtime, we include the calibration time with deep visual SLAM. For COLMAP's runtime, we exclude the calibration time, which is around 5 times longer than the current runtime. We will add more analysis on the efficiency of our initialization strategy, with the metrics listed in question 1.
>
> Finally, we want to express our sincere gratitude. Our paper will definitely be better with your suggestions. Thank you!
>
> -----------------------------
> ## **Reference:**
> [1] Jianing Yang, Alexander Sax, Kevin J. Liang et al. *Fast3R: Towards 3D Reconstruction of 1000+ Images in One Forward Pass*
>
> [2] Jianyuan Wang, Minghao Chen, Nikita Karaev et al. *VGGT: Visual Geometry Grounded Transformer*
>
> [3] Zhengqi Li, Richard Tucker, Forrester Cole et al. *MegaSaM: Accurate, Fast, and Robust Structure and Motion from Casual Dynamic Videos*
>
> [4] Jiakai Sun, Han Jiao, Guangyuan Li et al. *3DGStream: On-the-Fly Training of 3D Gaussians for Efficient Streaming of Photo-Realistic Free-Viewpoint Videos*

---

> > ### Comment · Reviewer_EE6C · 2025-08-04
> >
> > Dear Authors,
> >
> > My questions are addressed. Thank you for the clear response.

---

### Official Review · Reviewer_vtZE · 2025-06-30

**Clarity:** 2
**Significance:** 2
**Originality:** 2
**Rating:** 4
**Confidence:** 4

**Summary:**

This paper presents Instant4D, a pipeline for view synthesis from casual monocular videos. Compared to the baseline, the proposed approach makes significant modifications in initialization and covariance matrix parameterization. Specifically, it leverages grid-pruned colored point cloud derived from dense depth map to initialize the position and color of Gaussian primitives, and set different temporal scales for Gaussians belonging to static and dynamic regions. To reduce the number of parameters, the Gaussians are constrained to be axis-aligned, while deprecating view-dependent appearance. Experiment results demonstrate that the proposed pipeline achieves improved performance on multiple monocular nvs benchmarks while substantially increasing efficiency and compactness compared to the baseline.

**Questions:**

See weaknesses.

**Ethical Concerns:**

["NO or VERY MINOR ethics concerns only"]

**Final Justification:**

During the rebuttal, the authors did not provide any substantial new results to dispel my concerns about its novelty, methodology, and capability of novel view synthesis. In fact, their responses tend to avoid directly response to the issues and instead obscure it with some subtle expression.
For instance, in response to the concern that “all provided videos seem rendered from training viewpoints,”, they never explicitly confirm or refuted this point in the initial rebuttal, instead offer an indirect statement “For the visualization, we have addressed this issue by adding camera movement.”
And in the latest response, they try to distract with “outputs in the paper are rendered from novel viewpoints.”, but it irrelevant with the discussed video. I note that the videos in the website have been updated. But this does not invalidate the original critique. The original videos, which not can be found in the supplementary material, are obviously rendered from training camera trajectories.
A similar pattern also appears in their defense of the methodology.  Their latest response seems to implicitly confirm, via “under the same setting”, that the “Anisotropic“ in setting "w/ Anisotropic Gaussian" only describes the spatial dimension. Therefore, repeatedly citing the same unrelated result does not help address my concerns.
Even taking into account the updated videos and new comparison with suggested baseline provided in the discussion period, I still do not think this work significantly exceeds the acceptance threshold.

**Limitations:**

yes

**Paper Formatting Concerns:**

Not found formatting issues.

**Quality:**

2

**Strengths And Weaknesses:**

**Strengths**:

1.	The content is well-structured and easy to follow.

2.	Most adopted components are reasonable and exhibit clear improvements in ablations.

3.	The proposed method is evaluated on multiple monocular video datasets as well as in-the-wild AI generated videos.

**Weaknesses**:

1.	The ability for novel view synthesis lacks rigorous evaluation. All provided videos seems rendered from training viewpoints. The modification of train/test split for the quantitative evaluation on DyCheck dataset somewhat trivialize the task. It is recommended to provide comparison with more recent methods focused on monocular dynamic reconstruction, such as MoSca and Shape-of-Motion.

2.	The proposed simplified representation with a diagonal covariance matrix can be regarded as a "4D Gaussian Marble". But in the 4D case, such simplification eliminates the important capability of single Gaussian primitive for modeling motion stand-alone. Actually, the videos shown in supplement exhibit obvious flickering, it appears to stem from this constraint.

3.	The proposed framework lacks substantial technical novelty and essentially a natural integration of several existing modules.

4.	Typos: L154, 215, and 216 "NVIIDIA" ->"NVIDIA"; L52 “Datast” -> “Dataset”

---

> ### Author Rebuttal · Authors · 2025-07-28
>
> Thank you for your in-depth discussions and suggestions for our paper
>
> ## **Weakness:**
> ### **1: Qualitative Evaluation & Comparison with Other Methods**
> For the visualization, we have addressed this issue by adding camera movement. We promise to update the visualization part on our official website very soon after the reviewer discussion (following the rebuttal rules). For the train/test split, we believe Gaussian Splatting/NeRF was originally designed for interpolation between different viewpoints, therefore these per-scene methods naturally do not support extrapolation with large camera movement. Our train/test splitting could be viewed as temporal interpolation.
>
> For comparison with other methods, Shape-of-Motion [4] requires an additional tracking model to optimize, and the preprocessed file will reach 150 GB even for a 10-second video. Shape-of-Motion [4] also requires users to segment the mask themselves. Our method is fully automated. MoSca [5] utilizes depth, point track, and optical flow to reconstruct the scene and camera parameters. However, this design significantly increases the training time because of the complexity of the optimization. Due to time constraints, we are unable to present a complete evaluation. In the future, we will continue experiments on that benchmark.
>
> ### **2: Covariance Design**
> We choose to simplify the scaling matrix into one shared shape scale $s_{xyz}$ and one temporal scale $s_t$. We find that the isotropic design regularizes optimization and improves rendering quality as shown in Table 4.
>
> | Component               | PSNR ↑ | SSIM ↑ |
> |------------------------|--------|--------|
> | Motion Aware Gaussian  | 21.11  | 0.721  |
> | Isotropic Gaussian     | 23.00  | 0.661  |
> | SH                     | 23.52  | 0.755  |
> | **Ours (Full)**        | **24.52** | **0.834** |
>
>
> From our observations, this design can reduce overfitting and improve rendering quality from novel views. We assume the flickering behavior is caused by the lighting effects and texture changes in the bear's fur. We also tried the anisotropic Gaussians formulation, which does not solve this flickering behavior. Based on the overall trade-off, the simplified design is feasible. We will add this comparison for more visualization.
>
> ### **3: Technical Contribution**
> Our motivation is rooted in the recent advances in deep visual SLAM systems, which have significantly improved speed [1], robustness [2], and generalization [3] in geometry recovery from casually captured monocular videos. Traditional 4DGS methods initialize from the sparse point cloud calibrated from COLMAP. Our methods show that in the casual video setting, the training speed will be greatly improved by introducing our initialization strategy.
>
> Our work is, to our knowledge, one of the first to tightly integrate deep visual SLAM with 4D Gaussian Splatting, bridging the gap between geometry-centric SLAM research and appearance-centric neural rendering. We fully leverage the deep visual SLAM tool, achieving fast, robust, and efficient 4D reconstruction in monocular settings. We believe this direction is valuable for the community, as it provides a strong baseline, enables real-time rendering, and could be used as a dataset to train feedforward novel view synthesis models for casual video. We will open-source our code to further facilitate our research community.
>
> ### **4: Typo**
> We have corrected them in our manuscript. We will do a second proofreading of our manuscript.
>
> Finally, we sincerely thank you for your questions. Your insights have indeed helped us further refine our paper.
>
> -----------------------------
> ## **Reference:**
> [1] Jianing Yang, Alexander Sax, Kevin J. Liang et al. *Fast3R: Towards 3D Reconstruction of 1000+ Images in One Forward Pass*
>
> [2] Jianyuan Wang, Minghao Chen, Nikita Karaev et al. *VGGT: Visual Geometry Grounded Transformer*
>
> [3] Zhengqi Li, Richard Tucker, Forrester Cole et al. *MegaSaM: Accurate, Fast, and Robust Structure and Motion from Casual Dynamic Videos*
>
> [4] Qianqian Wang, Vickie Ye and Hang Gao et al. *Shape of Motion: 4D Reconstruction from a Single Video*
>
> [5] Jiahui Lei, Yijia Weng, Adam Harley et al. *MoSca: Dynamic Gaussian Fusion from Casual Videos via 4D Motion Scaffolds*

---

> > ### Comment · Reviewer_vtZE · 2025-08-03
> >
> > 1.	The main concern remains that the novel view synthesis capability has not been adequately evaluated. The author’s response appears to confirm that all presented videos are rendered from training viewpoints. Additionally, the defense of using its customized split in quantitative comparison is not convincing, and still no comparison is provided with more recent counterparts under the same settings.
> > 2.	Regarding the covariance design, please specify which part of the covariance is "anisotropic" in the second row of Table 4.  Since now the "isotropic" only describes the spatial dimension in current formulation (Since $s_{xyz}$ is obviously different with $s_{t}$), does it imply that the covariance between spatial and temporal dimensions remains 0 in the “anisotropic formulation”? If so, single Gaussian in this setting would still incapable of modeling motion. In my experience the flickering patterns observed in the presented video very likely resulted from the incapability of position-fixed Gaussian primitives to smoothly capture temporal transitions.
> > 3.	I do not think just using the outputs of MegaSAM for the optimization of Gaussian splatting can be called “tightly integration”.

---

> ### Author Response · Authors · 2025-08-04
> **Official Comment by Authors**
>
> We thank Reviewer vtZE for more detailed feedback, and we would love to respond further to these valuable comments.
>
> 1.
> * We would like to emphasize that all the presented outputs *in the paper* are rendered from novel viewpoints provided by the official evaluation sets. Additionally, we believe that there is a misunderstanding on the "rendered videos from training viewpoints". We hope to draw the attention of Reviewer vtZE around our rendered videos *on our website*. In fact, our rendering roughly followed the camera trace of the given video *but* we also rendered novel views for presentation in each video.
> * To further validate the effectiveness of our method, we evaluated it on the DyCheck dataset under *the same settings*, comparing it with the most recent approaches including MoSca [1], RoDyGS [2], Shape-of-Motion [3]):
> | Model    | Training Time (min)  | Rendering (FPS) | PSNR (dB) |
> |----------|----------------|---------------|---------|
> | MoSca  [1]  | 63  | 37  | **25.03** |
> | RoDyGS [2]  | *60*  | 67  |17.37|
> | Shape-of-Motion[3]|  74  | **131** | 23.84  |
> | **Ours**       | **11.2** |  *120* | *24.52* |
>
>     Note that:
>     - **Bold** denotes the best result in a column and *Italic* indicates the second best.
>     - For fair comparison, we implemented *all* these methods (including MoSca [1], which adopted ground-truth depth in the official code) using *estimated depth*. Training time includes preprocessing.
>
>     **Analysis:**
>     As shown in the table, our method reduces the training time by **over 80\%**, supporting the goal of fast casual-video reconstruction. Simultaneously, it remains competitive -- ranking second in terms of both rendering speed and quality.
>
>
> 2.
>
> * As Reviewer vtZE discussed, the design of "Isotropic Gaussian" degrades the center of primitive as: $\mu_{xyz|t}=\mu_{1:3}$ . Obviously, timestamp $t$ cannot affect the spatial center $\mu_{xyz}$. However, we argue that, modeling the relationship between opacity and timestamp by Equation $o_t = o \times \mathcal{N}(t, \mu_4, \Sigma_{4,4})$, along with more primitives kept by an aggressive grid pruning strategy, helps capture the motion information both *temporally* and *spatially*.
> * Under the same settings, the empirical results (refer to Table 4) prove the effectiveness of our design, as below:
>
> | Component               | PSNR ↑ | SSIM ↑ |
> |------------------------|--------|--------|
> | w/ Isotropic Gaussian        | **24.52** | **0.834** |
> | w/ Anisotropic Gaussian     | 23.00  | 0.661  |
>
> 3.
> * As shown in Figure 2, we apply the output of Mega-SAM [5] to the Geometric Recovery process. As mentioned by Reviewer vtZE, this process performs back-projection using the output camera parameters and video depth.
> * However, instead of a *naive addition*, we further explore the mechanism of Mega-SAM [5] and fuse its intermediate probability map $\hat{m} \in \mathbb{R}^{\frac{H}{8} \times \frac{W}{8}}$ into our framework. Specifically, we convert probablity map into motion mask to determine whether each Gaussian primitive is from static or dynmaic element and to assign suitable values of s_t. This ensures to initialize compact static background and dynamic objects with less overlappings. The empirical results prove the effectiveness of this fusion:
> | Component      | PSNR ↑ | SSIM ↑ |
> |------------------------|--------|--------|
> | w/ Motion Aware Gaussian      |   **24.52**     |   **0.834**  |
> | w/o Motion Aware Gaussian     |     21.11      |      0.721      |
>
>
>
> * For more clear illustration to readers, we will improve Figure 2 by describing how probability map flows in the Geometric Recovery process and the pipeline.
>
>
>
>
> -----------------------------
> ## **Reference:**
> [1] Jiahui Lei, Yijia Weng, Adam Harley et al. *MoSca: Dynamic Gaussian Fusion from Casual Videos via 4D Motion Scaffolds*
>
> [2] Yoonwoo Jeong, Junmyeong Lee, Hoseung Choi et al. *RoDyGS: Robust Dynamic Gaussian Splatting for Casual Videos*
>
> [3] Qianqian Wang, Vickie Ye and Hang Gao et al. *Shape of Motion: 4D Reconstruction from a Single Video*
>
> [4] Zeyu Yang, Hongye Yang, Zijie Pan *et al.* *Real-time Photorealistic Dynamic Scene Representation and Rendering with 4D Gaussian Splatting*.
>
> [5] Zhengqi Li, Richard Tucker, Forrester Cole et al. *MegaSaM: Accurate, Fast, and Robust Structure and Motion from Casual Dynamic Videos*

---

### Official Review · Reviewer_wosD · 2025-07-03

**Clarity:** 3
**Significance:** 3
**Originality:** 2
**Rating:** 4
**Confidence:** 3

**Summary:**

This paper presents a 4D reconstruction method that offers fast training speeds and good reconstruction fidelity.

Leveraging the MegaSAM and Unidepth methods, Instant4D initializes the Gaussian primitives through backprojection and voxel pruning. These initializations are then fed into the optimization process of 4DGS, which incorporates a separate temporal dimension in the properties of the 3D Gaussian primitives.

During optimization, Instant4D replaces high-order SCH with direct RGB values, uses isotropic Gaussians, and applies higher temporal scaling factors for static regions. The static regions are inferred from MegaSAM's predictions. These techniques contribute to improved reconstruction quality on the NVIDIA dataset.

Experiments conducted on both the NVIDIA and Dycheck datasets demonstrate that the proposed method not only accelerates the training process but also yields superior reconstruction quality on these two datasets.

**Questions:**

My main concerns stem from the insufficient experimental validation presented in this paper:

1. This method relies solely on the Visual SLAM calibration approach, which makes it unfair to directly compare with mainstream methods that use COLMAP calibration.
2. The paper lacks experiments on the Neu3D dataset.

If the authors can provide more comprehensive experimental results that demonstrate the robust performance of the proposed techniques, I would be inclined to raise my score.

**Ethical Concerns:**

["NO or VERY MINOR ethics concerns only"]

**Final Justification:**

Many thanks to the authors for their response regarding the main contribution. My concerns have been well addressed, and the rebuttal has clarified the method's contributions—specifically, the integration of deep visual SLAM into 4DGS reconstruction, the pruning of initial point clouds, and the introduction of the isotropic Gaussian formulation. The experimental results on the multi-camera Neu3D dataset further demonstrate the effectiveness of the proposed approach. I suggest that the authors include these details and experimental findings in the final version of the paper.

**Limitations:**

Yes

**Paper Formatting Concerns:**

No paper formatting concerns.

**Quality:**

3

**Strengths And Weaknesses:**

### Strengths:
1. The paper is well-written and easy to follow.
2. The initialization of Gaussian properties using MegaSAM and the Unidepth method is simple and straightforward.
3. The use of motion masks generated by MegaSAM helps separate dynamic foreground objects, improving the overall reconstruction quality.

### Weakness:
1. The aggregation process via the hash map in L151 is unclear and requires further explanation.
2. The paper omits a comparison with the work "4D Gaussian Splatting for Real-Time Dynamic Scene Rendering" [A].
3. The paper also omits experiments on the datasets proposed in "Neural 3D Video Synthesis from Multi-view Video" [B]. This omission weakens the overall validation of the proposed techniques, making their effectiveness less convincing.
4. Since the method aims to reduce training time, it would be beneficial if the authors could provide a detailed comparison of how the proposed techniques improve training speed relative to the baseline method, 4DGS.
5. The technical contributions in this paper are somewhat limited and lack significant innovation. From my perspective, the core contribution lies in the Gaussian initialization strategy that leverages existing visual SLAM and depth prediction methods. The voxel pruning strategy does not appear to offer novel insights.
6. As shown in Table 1, the authors differentiate between methods using COLMAP calibration and Visual SLAM calibration. Based on the performance comparison of the 4DGS method, COLMAP calibration seems to result in better reconstruction performance. This raises the question of whether it is worth sacrificing the accuracy of COLMAP calibration in favor of Visual SLAM calibration. It would be more beneficial if the method could combine the accuracy of COLMAP with the useful predictions from Visual SLAM.


#### Reference
A. Wu G, Yi T, Fang J, et al. 4d gaussian splatting for real-time dynamic scene rendering[C]//Proceedings of the IEEE/CVF conference on computer vision and pattern recognition. 2024: 20310-20320.
B. Li T, Slavcheva M, Zollhoefer M, et al. Neural 3d video synthesis from multi-view video[C]//Proceedings of the IEEE/CVF conference on computer vision and pattern recognition. 2022: 5521-5531.

---

> ### Author Rebuttal · Authors · 2025-07-28
>
> Thank you for sharing your precious advice for our paper. Our discussions of the weaknesses and concerns are listed below.
>
> ## **Weakness:**
> ### **1: More Details about the Grid Pruning Implementation**
>
> Our grid pruning implementation follows the voxel-based filtering approach in Open3D [1] and PointCloudUtils [2]. Specifically, each point is discretized into voxel coordinates (via integer division by voxel size) and inserted into a hash map using these coordinates as keys within constant time complexity. Points within the same voxel are grouped, averaging their positions and attributes (RGB, covariance) into one representative Gaussian. We will clarify this implementation explicitly in our revision.
>
>
> ### **2: Comparison with "4D Gaussian Splatting for Real-Time Dynamic Scene Rendering" [A]**
>
> The comparisons to "4D Gaussian Splatting" [A] have been presented in our manuscript, denoted as "4DGS [31]" in Table 1 and Table 3.
>
>
> **On the NVIDIA dataset (Table 1):**
>
> | Method     | Calibration  | Runtime ↓ | FPS ↑ (480p) | FPS ↑ (860p) | PSNR ↑ |
> |------------|--------------|-----------|--------------|--------------|--------|
> | 4DGS [A]   | COLMAP       | 1.2 h     | 43           | 29           | 21.45  |
> | **Ours**   | Visual SLAM  | **0.02 h**| **822**      | **676**      | **23.99** |
>
> **On the Dycheck dataset (Table 3):**
>
> | Method      | Apple | Block | Paper | Spin | Teddy | Avg PSNR ↑ | Runtime ↓ (h) | Mem ↓ (GB) |
> |-------------|-------|-------|-------|------|-------|------------|---------------|-------------|
> | 4DGS [A]    | 23.24 | 22.05 | 21.03 | 22.99| 18.89 | 21.64      | 1.2           | 21          |
> | Ours (Lite) | 24.90 | 23.48 | 23.18 | 23.60| 19.96 | 23.02      | **0.03**      | **1.1**     |
> | Ours (Full) | **26.84** | **23.98** | **24.77** | **25.25** | **21.78** | **24.52** | 0.12 | 8 |
>
> We apologize for the confusion regarding the method naming. For clarity, we will rename the 4DGS [37] as "Real-time 4DGS" in the camera-ready version to avoid ambiguity.
>
> ### **3: Experiment on the Neu3D dataset.**
> We conducted experiments on Neu3D below:
>
> | Methods       | Coffee Martini | Spinach | Cut Beef | Flame Steak | Sear Steak | Overall |
> |---------------|----------------|---------|----------|-------------|------------|---------|
> | MixVoxels [7] | 29.36          | 31.61   | 31.30    | 31.21       | 31.43      | 30.98   |
> | NeRFPlayer [8]| 31.53          | 30.56   | 29.35    | 31.93       | 29.12      | 30.50   |
> | HyperReel [9] | 28.37          | 32.30   | 32.92    | 32.20       | 32.57      | 31.67   |
> | 4DGS [10]     | 28.33          | 32.93   | 33.85    | 34.03       | 33.51      | 32.53   |
> | **Ours**      | 27.53          | 33.18   | 31.02    | 33.89       | 28.25      | 30.77   |
> ---
> The primary focus of our work is the reconstruction of scenes from *casual monocular* videos, characterized by a single camera undergoing free movement. The Neu3D [6] dataset, however, features static *multi-camera* setups *without camera movement*. Our method tries to handle a general set of casual videos that does not require synchronized multi-view videos.
> Thus, some of our designs like initialization strategy and Gaussian simplification formulation are incompatible with Neu3D's static multi-camera setup [6]. However, the performance of our method is still on-par with the most competitive methods.
>
> We will also add this evaluation in the supplementary material for our camera-ready version.
>
> ### **4: Detailed Comparison on how the Proposed Techniques Improve Training Speed Relative to the Baseline method**
>
> We do analyze how these proposed techniques improve training speed through the ablation study in Table 2. Following your suggestion, we now refactor this table as shown below for better clarity. The two major techniques related to the training speed are grid pruning and the color design.
>
> In the table below, the Baseline method shares similar methods with Real-time 4DGS. "+Pruning" method denotes that instead of starting from the sparse point cloud calibrated from COLMAP, we apply our initialization strategy without requiring densification. As shown in the table, one benefit is that all Gaussians are present at the beginning and can immediately receive gradient updates. The second change is the trade-off for spherical harmonics function design. In this ablation study, experiments show that, given dense point cloud initialization, direct RGB color is sufficient to reconstruct monocular dynamic scenes. It is also worth noting that a third-order spherical harmonics function requires **48** parameters. In comparison, other parameters e.g. (position, rotation, scaling) take **13** parameters in total.
>
>
> Updated Ablation Study Table 2:
> | Model        | PSNR ↑ | Runtime (sec) | Memory (MB) | Rendering FPS |
> |--------------|--------|---------------|-------------|---------------|
> | Baseline     | 23.38  | 181           | 10676       | 154           |
> | + Pruning    | 23.83  | 59            | 2806        | 588           |
> | - SH         | **23.99** | **42**     | **832**     | **981**       |
>
>
> - Our final version applies both of these techniques and thus achieves 4x less runtime and 13x less memory requirement, compared with the baseline method 4DGS.
> - We will follow your suggestions, update our Table 2 in the camera ready, and include a detailed, reader-friendly analysis on how the proposed techniques improve training speed relative to the baseline method.
>
>
> ### **5: Discussion on the contribution of our work.**
>
> Our motivation is rooted in the recent advances in deep visual SLAM systems, which have significantly improved speed [3], robustness [4] and generalization [5] in geometry recovery from casually captured monocular videos. Traditional 4DGS methods initialize from the sparse point cloud calibrated from COLMAP. From the deep visual SLAM system, we are able to acquire the point cloud sequence. However, we find the redundant gaussians greatly slow the overall training speed if we directly input the point cloud sequence into our optimization pipeline. To solve this problem, we apply a simple yet effective grid pruning method, which reduces the number of Gaussians by 92%. Experiments show that our method achieves a 30× speed-up and our method is the first to reduce optimization time for a casual video to minute-level, facilitating numerous downstream applications.
>
> To further overcome challenges posed in the monocular setting, we also introduce an isotropic gaussian formulation, regularizing the expression and increasing the rendering quality by **1.52 dB**. To solve the problem that the gaussians outside of the frustum tend to disappear, we design the motion-aware gaussian strategy, improving the render result by **3.41 dB** PSNR compared with the baseline. Experiments show that our method achieves a **30×** speed-up and reduces training time to within two minutes, facilitating numerous downstream applications.
>
>
> ### **6: Discussion on the COLMAP calibration and Visual SLAM calibration.**
>
> COLMAP is designed for *static* scenes from *consistent* videos. However, it commonly fails in *dynamic and casually captured* videos. Visual SLAM systems, in contrast, are more robust to motion blurring [4], fast to optimize [5] or feedforward [4] and avoid manual masking.
>
> We provide the evaluation with COLMAP as calibration as shown below.
>
> | Method      | Calibration | Runtime (s) | FPS (480p) | PSNR ↑ |
> |-------------|-------------|-------------|------------|-----------|
> | Ours-COLMAP | COLMAP      | 252 s       | 103        | 23.45     |
> | **Ours**    | Visual SLAM | 72 s        | **822**    | **23.99** |
> ----
>
>
> We also considered combining Visual SLAM's prediction with COLMAP's pose estimates. However, such pairing is rarely available in practice, as these systems operate independently and rely on different camera intrinsics, scales, and assumptions. WWe will clarify the grid pruning implementation explicitly in the revised manuscript, and we appreciate the opportunity to explain this more clearly.
>
>
> ## **Question:**
>
> ### **1. Comparison with COLMAP calibration**
> COLMAP often outputs highly accurate camera poses, but at the same time requires more than 15 minutes optimization. When evaluating, the common practice is to use the COLMAP camera pose as ground truth. Therefore we argue that it is fair to compare with mainstream methods. It is also worth noting that for the runtime reported for COLMAP methods, we do not include the calibration time, while we include the calibration time in our method.
>
> ### **2. Evaluation on the Neu3D Dataset**
> Please refer to our discussion in weakness 3.
>
>
> -----------------------------
> ## **Reference**
>
> [1] Qian‑Yi Zhou, Jaesik Park, and Vladlen Koltun. *Open3D: A Modern Library for 3D Data Processing*.
>
> [2] Francis Williams. *Point Cloud Utils*
>
> [3] Jianing Yang, Alexander Sax, Kevin J. Liang et al. *Fast3R: Towards 3D Reconstruction of 1000+ Images in One Forward Pass*
>
> [4] Jianyuan Wang, Minghao Chen, Nikita Karaev et al. *VGGT: Visual Geometry Grounded Transformer*
>
> [5] Zhengqi Li, Richard Tucker, Forrester Cole et al. *MegaSaM: Accurate, Fast, and Robust Structure and Motion from Casual Dynamic Videos*
>
> [6] Tianye Li, Mira Slavcheva, Michael Zollhoefer et al. *Neural 3D Video Synthesis from Multi-view Video*
>
> [7] Feng Wang, Sinan Tan, Xinghang Li, Zeyue Tian et al. *Mixed Neural Voxels for Fast Multi-view Video Synthesis*.
>
> [8] Liangchen Song, Anpei Chen, Zhong Li et al. *NeRFPlayer: A Streamable Dynamic Scene Representation with Decomposed Neural Radiance Fields*.
>
> [9] Benjamin Attal, Jia-Bin Huang, Christian Richardt et al. *HyperReel: High-Fidelity 6-DoF Video with Ray-Conditioned Sampling*.
>
> [10] Zeyu Yang, Hongye Yang, Zijie Pan *et al.* *Real-time Photorealistic Dynamic Scene Representation and Rendering with 4D Gaussian Splatting*.

---

> ### Comment · Reviewer_wosD · 2025-08-07
>
> Many thanks to the authors for their response regarding the main contribution. My concerns have been well addressed, and the rebuttal has clarified the method's contributions—specifically, the integration of deep visual SLAM into 4DGS reconstruction, the pruning of initial point clouds, and the introduction of the isotropic Gaussian formulation. The experimental results on the multi-camera Neu3D dataset further demonstrate the effectiveness of the proposed approach. I suggest that the authors include these details and experimental findings in the final version of the paper.
>
> Additionally, I encourage the authors to provide a more thorough description of the implementation details for the Neu3D experiments, given that the method is originally designed for monocular videos.

---

> > ### Author Response · Authors · 2025-08-07
> > **Official Comment by Authors**
> >
> > We sincerely thank reviewer wosD for the valuable suggestions. We will incorporate the updated table, technical details, and Neu3D experimental results into the final version of the paper.
> >
> > Regarding the Neu3D experimental setup, we made minimal modifications to adapt our method to the multi-camera video setting. Specifically, we initialize from a sparse point cloud generated via COLMAP calibration and activate the densification module. To ensure appropriate fitting for the multi-view setting, we disable the isotropic Gaussian design. All other components remain consistent with those in our monocular setting. We will include these implementation details in the final version to ensure clarity and reproducibility.

---

> > > ### Comment · Reviewer_wosD · 2025-08-09
> > >
> > > Thanks for the detailed response. Most of my concerns have been addressed and I'd like to raise my rating score.

---

### Official Review · Reviewer_K6kc · 2025-07-03

**Clarity:** 2
**Significance:** 3
**Originality:** 4
**Rating:** 5
**Confidence:** 4

**Summary:**

This paper proposes a system for efficiently compressing and accelerating precise reconstruction of scenes and moving objects in dynamic environments. The pipeline first performs initialization by separately optimizing camera intrinsics, extrinsics, and depth, simplifying the optimization function to avoid local minima and the computational burden of numerous parameters. Subsequently, it identifies moving objects to allocate more computational resources to them and employs voxel grid pruning to prevent overfitting. Finally, the paper simplifies the optimization formula for 4D Gaussians and increases the temporal scale of the static background to maintain the consistency and stability of the optimization algorithm. The main contributions of this work lie in the initialization with state-of-the-art techniques, a novel pruning strategy, and redundancy reduction in existing methods.

**Questions:**

Have you attempted to significantly zoom in and out on the constructed 4DGS map to evaluate whether the pruning method maintains good rendering performance for new views?

For input views with motion blur or defocus blur, have you applied any special processing?

Have you tried depth prediction methods based on video?

**Ethical Concerns:**

["NO or VERY MINOR ethics concerns only"]

**Final Justification:**

Overall, this is a good paper with solid contributions which merit publication.
The authors addressed the reviewers concerns well in the rebuttal. Therefore, I would raise the score to 5.

**Limitations:**

The paper effectively discusses its limitations but does not clearly address the robustness to motion blur or defocus blur.

**Paper Formatting Concerns:**

No major formatting issues.

**Quality:**

2

**Strengths And Weaknesses:**

Paper Strengths:

1. The paper proposes a targeted pruning algorithm that reduces the number of scene Gaussians while maintaining accurate scene reconstruction.

2. The paper identifies redundancies in existing works and simplifies the problem setting to maintain consistent performance while accelerating inference speed and reducing model size.

3. The paper proposes a pipeline for depth and camera intrinsic/extrinsic parameter estimation that leverages state-of-the-art techniques.

Major Weaknesses:
1. The running time data directly taken from other papers is not very convincing, as these works may not have been run on the same GPU or CPU.

2. The paper only compares different methods based on the same calibration estimation, without designing performance comparisons under the same initial depth, such as depth optimized via the same multi-view approach. This suggests that performance improvements may stem from advancements in the underlying depth estimation algorithm.

3. There is no comparison between the optimized depth and the actual depth, and first-order optimization may be insufficient for ensuring depth consistency.

4. The method is not evaluated under different depth settings, leaving uncertainty about whether the performance improvements are solely dependent on a specific depth estimation algorithm.

Minor Weaknesses:
- The visual comparisons on the Dycheck dataset appear to be observed from different viewpoints, with positional deviations. The 4D GS and the proposed method (Ours) share the same viewpoint, but the Ground Truth appears to be slightly shifted to the right.

- The sentence "Those dynamic Gaussians, will change there position and scaling according to the deviation of timestamp" contains a serious spelling error, where "there" should be "their."

---

> ### Author Rebuttal · Authors · 2025-07-28
>
> Thank you for your in-depth discussions and constructive suggestions for our paper. Our responses to the weaknesses and questions are listed below.
>
> ## **Major Weakness:**
> ### **1: Runtime Analysis with Other Work**
> Before clarification, we show a "detailed" Table 1 for the comparison of efficiency:
>
> | Method         | Runtime (h) | Rendering FPS (480p) |  Compute |
> |----------------|-------------|----------------------|----------|
> | *Official Results* |
> | HyperNeRF [1]     | 64          | 0.40              | 4 × TPU v4s |
> | DynamicNeRF [2]   | 74          | 0.05              | 1 × V100 |
> | RoDynRF [3]       | 28          | 0.42              | 1 × V100 |
> | 4DGS [4]          | 1.2         | 43                | 1 × RTX 3090 |
> | ~Casual-FVS [5]~     | 0.25        | 48               | 1 × A100 |
> | *Our implementations* |
> | InstantSplat [6]  | 0.15        | 117              | 1 × A6000 |
> | 4DGS [7]         | 0.16        | 98                | 1 × A6000 |
> | Ours           | **0.02**    | **822**              | 1 × A6000 |
> ---
> * For fair comparison, we implement a popular and competitive method InstantSplat [6] and the baseline 4DGS [7] and compare our method with them using *the same server settings*.
> * The implementation of Casual-FVS [5] has not been open-sourced, so we can hardly report the runtime with the same server settings. We will not compare our method with it in Table 1 for our camera-ready version, but we will keep citing it.
> * Our method performs over 1000x faster than some previous NeRF methods like HyperNeRF [1], DynamicNeRF [2] and RoDynRF [3]. Though these methods used different GPUs from ours, we believe the obvious efficiency superiority of our method can show its competitiveness.
>
>
> ### **2: Discussion on Depth Estimation in our Work**
> For a detailed discussion, we list the updated Table 4 below:
>
> | Component               | PSNR ↑ | SSIM ↑ |
> |------------------------|--------|--------|
> | **Ours (Full)**        | **24.52** | **0.834** |
> ||
> | w/o Motion Aware Gaussian  | 21.11  | 0.721  |
> | w/o Isotropic Gaussian     | 23.00  | 0.661  |
> | w/o zero-degree SH                     | 23.52  | 0.755  |
>
> ---
> We acknowledge that the speedup and the performance increase are related to the underlying depth estimation algorithm, while we believe our contributions such as the pruning strategy and simplified 4D Gaussian representation are orthogonal to the specifics of the initial depth estimation.
>
> Our ablation study shows that the isotropic Gaussian formulation regularizes the training and increases the rendering quality by 1.52 dB. To solve the problem that the Gaussians outside of the frustum tend to disappear, we design the motion-aware Gaussian strategy, improving the render result by 3.41 dB PSNR compared with the baseline. Experiments show that our method achieves a 30× speed-up and reduces training time to within two minutes. As a result, our improvement does not merely stem from the depth estimation.
>
> ### **3: Comparison between the Optimized Depth and the Actual Depth**
> We conducted a new experiment on the Dycheck dataset to compare the depth estimated by Unidepth [8] with the ground-truth depth. We will also incorporate the table below in our supplementary material for our camera-ready version.
>
> | Method         | Abs-Rel $\downarrow$|  Log-RMSE $\downarrow$ | $\delta_{1.25}$ $\uparrow$|
> |----------------|------------------|-------------------|-------------------|
> | Unidepth[8] | 0.254       | 0.303         | 68.5          |
> | Optimized   | 0.137        | 0.254          | 89.6         |
>
> The comparison shows that the first-order optimization reduces error in Abs-Rel by 46.1% and in Log-RMSE by 16.2%, and increases $\delta_{1.25}$ accuracy by 30.8%.
>
> ### **4: Evaluation under Different Depth Setting**
> Thank you for emphasizing the importance of depth estimation. As shown in DepthSplat [9], better depth leads to improved novel view synthesis quality. The current evaluations are done under short real-world video and AI generated videos that are featured with foreground moving objects and static background, which does not involve different depth settings. We will add more discussion about the depth setting.
>
> ## **Minor Weakness:**
> ### **1 & 2**
> We have fixed this problem and will do a thorough proofreading.
>
> ## **Question:**
> ### **1: Zoom-in & Zoom-out Effect**
> With significant zoom-in, we find that our method tends to display clear boundaries without outlier Gaussians that are frequent in 3D Gaussian Splatting and other dynamic reconstruction work. We do not find unusual visual effects other than this. With significant zoom-in, we observe some voidness between Gaussians. This is not observed in the baseline methods.
>
> ### **2: Deblurring for Motion and Defocus**
> Motion blurring and defocus blurring are challenging problems. These problems are not quite obvious when evaluating since we typically have a higher video frame rate compared with the camera and object motion. While current dynamic methods don't focus on this problem, we are open to exploring this issue.
>
> ### **3: Application with Video Depth Model?**
> We agree that better depth estimation will improve the result. However, we find that a video-depth prediction model might not be the best solution in our method. For instance, the NVIDIA dynamic dataset is featured with a 12-camera teleporting setup, which is incompatible with the assumptions in the video depth model.
>
> Finally, we sincerely thank you for your questions. Your insights have indeed helped us further refine our paper.
>
> -----------------------------
> ## **Reference**
>
> [1] Keunhong Park, Utkarsh Sinha, Peter Hedman *et al.* *HyperNeRF: A Higher-Dimensional Representation for Topologically Varying Neural Radiance Fields*.
>
> [2] Chen Gao, Ayush Saraf, Johannes Kopf *et al.* *Dynamic View Synthesis from Dynamic Monocular Video*.
>
> [3] Yu-Lun Liu, Chen Gao, Andreas Meuleman *et al.* *Robust Dynamic Radiance Fields*.
>
> [4] Guanjun Wu, Taoran Yi, Jiemin Fang *et al.* *4D Gaussian Splatting for Real-Time Dynamic Scene Rendering*.
>
> [5] Yao-Chih Lee, Zhoutong Zhang, Kevin Blackburn-Matzen *et al.* *Fast View Synthesis of Casual Videos with Soup-of-Planes*.
>
> [6] Zhiwen Fan, Wenyan Cong, Kairun Wen *et al.* *InstantSplat: Sparse-view Gaussian Splatting in Seconds*.
>
> [7] Zeyu Yang, Hongye Yang, Zijie Pan *et al.* *Real-time Photorealistic Dynamic Scene Representation and Rendering with 4D Gaussian Splatting*.
>
> [8] Luigi Piccinelli, Yung-Hsu Yang, Christos Sakaridis *et al.* *UniDepth: Universal Monocular Metric Depth Estimation*.
>
> [9] Haofei Xu, Songyou Peng, Fangjinhua Wang *et al.* *DepthSplat: Connecting Gaussian Splatting and Depth*.

---

> > ### Comment · Reviewer_K6kc · 2025-08-08
> >
> > I thank the authors for their experiments and clarifications. I have also reviewed other comments and the authors' responses. The authors' thorough experiments on runtime sufficiently demonstrate the significant advantages of this work in terms of runtime and memory. However, the impact of depth-dependent modules on performance may still require further experiments to explore.
> > For instance, whether motion-aware Gaussians require sharper depth boundaries is likely strongly related to the underlying depth estimator, especially in cases where moving objects are very small in the view and occlude each other. Due to Mega-SAM's low-resolution optimization, it may not handle these issues effectively, potentially generating incorrect masks that affect the motion-aware module.
> > UniDepth itself exhibits inconsistent performance in large-scale scenes. For example, in UDGS-SLAM [A], larger office scenes show relatively poor pose estimation, which is closely tied to its inaccurate depth estimation. Does your first-order optimization address this issue to prevent the system from overfitting to specific scenes? Comparing averages with ground-truth depth may not accurately reflect inconsistencies across multi-frame depth estimations. Can you demonstrate that your contributions are applicable and beneficial with other depth estimation methods, and that they can leverage improved depth estimation quality rather than overfitting to the current depth estimator? This would be a more significant contribution to the field.
> > Indeed, NeRF- and GS-based methods have limited extrapolation capabilities. However, some qualitative evaluation is still needed for viewpoints beyond those specified in official datasets, such as extreme zooming in or out. The authors mention that extreme zooming only revealed voidness between Gaussians due to excessive pruning, suggesting some extrapolation capability. I hope future versions of the paper can elaborate on the comparison of voidness between Gaussians across different scenes and with other methods.
> > By utilizing pruning and generating motion masks with Mega-SAM to segment dynamic objects, the authors simplify existing state-of-the-art methods to achieve excellent real-time performance and competitive rendering quality. While these methods may lack some novelty, designing more rigorous experiments to demonstrate that the method maintains comparable performance to other advanced methods under pruning across diverse scenes and inputs would be a significant contribution to the community.
> > [A] Mansour, Mostafa, et al. "UDGS-SLAM: UniDepth assisted Gaussian splatting for monocular SLAM." Array (2025): 100400.

---

> ### Author Response · Authors · 2025-08-09
> **Official Comment by Author**
>
> We thank reviewer K6kc for the valuable feedback.
>
> ### 1: Depth-Dependent Module Analysis
> Originally, we applied MegaSAM [7] for the Geometric Recovery process, using UniDepth [3] as depth estimator and performing the consistent depth optimization (denoted as *Ours*).
> To assess whether our method overfits to a specific depth estimator, we disabled the consistent depth optimization and replaced the depth estimator UniDepth [3] with its alternatives VideoDepthAnything [4] and Cut3R [5], denoted as *Ours-VideoDepthAnything* and *Ours-Cut3R*, respectively. For reference, we also provide a UniDepth version without consistent depth optimization, denoted as *Ours-UniDepth*.
>
> The results on the Dycheck dataset are shown below:
>
> | Method         | PSNR $\uparrow$| SSIM $\uparrow$|
> |----------------|----------------|----------------|
> | RoDyGS [1]     | 17.37          | 0.46           |
> | RoDynRF [2]    | 14.90          | 0.39           |
> ||
> | Ours-UniDepth [3]          | 22.68          | 0.70           |
> | Ours-VideoDepthAnything [4] | 24.36          | 0.81           |
> | Ours-Cut3R [5]              | 24.18          | 0.80           |
> | **Ours** (original)                    | **24.52**      | **0.83**       |
> ---
> The results show that across various depth estimators, the proposed method consistently outperforms previous state-of-the-art methods, RoDynRF [1] and RoDyGS [2] by a large margin. This fact demostrates that the improvements of the proposed method are not tied to a specific depth estimator, and the method does not overfit to a particular depth setting.
>
>
> ### 2: Does your first-order optimization address this issue to prevent the system from overfitting to specific scenes?
>
> Our approach performs per-scene optimization to enable real-time rendering at inference. We do not expect the model to generalize depth predictions across unrelated scenes.
>
>
> ### 3: UniDepth itself exhibits inconsistent performance in large-scale scenes.
>
> In our deep visual SLAM part, we do not rely solely on Unidepth [3] to generate sufficiently accurate depth. Instead, we apply additional optimization to acquire consistent video depth, as shown in the following table.
>
> | Method         | Abs-Rel $\downarrow$|  Log-RSME $\downarrow$ | $\delta_{1.25}$ $\uparrow$|
> |----------------|------------------|-------------------|-------------------|
> | Unidepth[8] | 0.254       | 0.303         | 68.5          |
> | Optimized   | 0.137        | 0.254          | 89.6         |
>
>
> On the Dycheck dataset, the optimization reduces Abs-Rel and Log-RSME errors by **46.1%**, **16.2%** respectively, indicating that the refined depth can be quantitatively closer to the ground truth. Additionally, $Delta_{1.25}$ indicates that **89.6%** of pixels in the optimized depth are within ±25% of the ground-truth depth. This experiment showcases that although Unidepth produces scale shifts, our optimization yields a temporally stable depth aligned with the ground truth scene geometry.
>
>
> ### 4: Elaborate on the comparison of voidness between Gaussians across different scenes and with other methods.
>
> We thank reviewer K6kc for the suggestions, we will include more visualization and analysis around the zoom-in and zoom-out viewpoint across various scenes, compared with the baseline 4DGS [6], and previous state-of-the-art methods RoDyGS [2] and MoSca [8].
>
> ---
> [1] Yu-Lun Liu, Chen Gao, Andreas Meuleman *et al.* *Robust Dynamic Radiance Fields*.
>
> [2] Yoonwoo Jeong, Junmyeong Lee, Hoseung Choi et al. *RoDyGS: Robust Dynamic Gaussian Splatting for Casual Videos*
>
> [3] Luigi Piccinelli, Yung-Hsu Yang, Christos Sakaridis et al. *UniDepth: Universal Monocular Metric Depth Estimation*.
>
> [4] Sili Chen, Hengkai Guo, Shengnan Zhu et al. *Video Depth Anything: Consistent Depth Estimation for Super-Long Videos*
>
> [5] Qianqian Wang, Yifei Zhang, Aleksander Holynski et al. *Continuous 3D Perception Model with Persistent State*
>
> [6] Zeyu Yang, Hongye Yang, Zijie Pan *et al.* *Real-time Photorealistic Dynamic Scene Representation and Rendering with 4D Gaussian Splatting*.
>
> [7] Zhengqi Li, Richard Tucker, Forrester Cole et al. *MegaSaM: Accurate, Fast, and Robust Structure and Motion from Casual Dynamic Videos*
>
> [8] Jiahui Lei, Yijia Weng, Adam Harley et al. *MoSca: Dynamic Gaussian Fusion from Casual Videos via 4D Motion Scaffolds*

---

### Author Response · Authors · 2025-08-09
**Author's General Response**

We sincerely thank all the reviewers for their constructive feedback and thoughtful suggestions. We are encouraged by their recognition of:

1. Our method performs 4D reconstruction of *any casually captured video* within minutes, with a **30×** speedup and a **92%** reduction in memory usage compared to the state of the art (Reviewers K6kc, wosD, vtZE, EE6C).

2. We propose a simple yet effective grid pruning method, reducing the number of Gaussians to **8%**. We design a simplified 4D Gaussian Splatting pipeline tailored for a monocular setting. The design of both motion-aware masking and simplification of Gaussian primitives shows clear improvements in ablations (Reviewer vtZE), appears well-adaptable to different 4D Gaussian Splatting methods (Reviewer EE6C), and improves the overall reconstruction quality (Reviewer wosD).

3. Our method generalizes to multiple scenarios, such as monocular videos (e.g., NVIDIA dynamic & Dycheck iPhone) mentioned by Reviewer EE6C, multi-view videos (e.g., Neu3D) mentioned by Reviewer wosD, and SORA-generated videos (e.g., website examples) mentioned by Reviewer vtZE. Evaluations across these datasets further demonstrate its effectiveness (Reviewer wosD).

In response to the reviewers’ concerns, we have provided point-by-point, detailed (Reviewer wosD), and clear (Reviewer EE6C) replies and we have incorporated the corresponding revisions into the main paper.

---

### Decision · Program_Chairs · 2025-09-17

**Decision:**

Accept (poster)

**Comment:**

This paper got two accept and two borderline accept. One reviewer pointed out more experimental results are expected to further validate the proposed approach. Please include more experimental results in the final camera ready paper.